# Hypothyroidism and Heart Rate Variability: Implications for Cardiac Autonomic Regulation

**DOI:** 10.3390/diagnostics14121261

**Published:** 2024-06-14

**Authors:** Carina Bogdan, Viviana Mihaela Ivan, Adrian Apostol, Oana Elena Sandu, Felix-Mihai Maralescu, Daniel Florin Lighezan

**Affiliations:** 1Department VII, Internal Medicine II, Discipline of Cardiology, “Victor Babeş” University of Medicine and Pharmacy, Eftimie Murgu Sq. No. 2, 300041 Timişoara, Romania; carina.bogdan@umft.ro (C.B.); adrian.apostol@umft.ro (A.A.); oana.ciolpan@umft.ro (O.E.S.); 2Centre for Molecular Research in Nephrology and Vascular Disease, “Victor Babeş” University of Medicine and Pharmacy, 300041 Timișoara, Romania; mihai.maralescu@umft.ro; 3Department of Internal Medicine II, Discipline of Nephrology, “Victor Babeș” University of Medicine and Pharmacy, 300041 Timișoara, Romania; 4Department V, Internal Medicine I, Discipline of Medical Semiology I, “Victor Babeș” University of Medicine and Pharmacy, 300041 Timișoara, Romania; dlighezan@umft.ro; 5Center of Advanced Research in Cardiology and Hemostaseology, “Victor Babeș” University of Medicine and Pharmacy, E. Murgu Square, Nr. 2, 300041 Timișoara, Romania

**Keywords:** heart rate variability, autonomic dysfunction, hypothyroidism

## Abstract

Thyroid hormones have a pivotal role in controlling metabolic processes, cardiovascular function, and autonomic nervous system activity. Hypothyroidism, a prevalent endocrine illness marked by inadequate production of thyroid hormone, has been linked to different cardiovascular abnormalities, including alterations in heart rate variability (HRV). The study included 110 patients with hypothyroid disorder. Participants underwent clinical assessments, including thyroid function tests and HRV analysis. HRV, a measure of the variation in time intervals between heartbeats, serves as an indicator of autonomic nervous system activity and cardiovascular health. The HRV values were acquired using continuous 24-h electrocardiogram (ECG) monitoring in individuals with hypothyroidism, as well as after a treatment period of 3 months. All patients exhibited cardiovascular symptoms like palpitations or fatigue but showed no discernible cardiac pathology or other conditions associated with cardiac disease. The findings of our study demonstrate associations between hypothyroidism and alterations in heart rate variability (HRV) parameters. These results illustrate the possible influence of thyroid dysfunction on the regulation of cardiac autonomic function.

## 1. Introduction

The autonomic nervous system is a critical component of the peripheral nervous system that regulates involuntary physiological functions essential for maintaining homeostasis and responding to stress. It controls various involuntary processes, including cardiovascular regulation, respiratory rate, gastrointestinal motility, and glandular secretion [1,2].

Cardiac dysautonomia is the malfunction of the autonomic nervous system-regulated cardiovascular hemostasis. Dysautonomia’s impact on the cardiovascular system can result in a diverse array of symptoms and complications and can have a negative effect on the health of a person, leading to an elevated risk of cardiovascular disease [3].

Hypothyroidism occurs when the thyroid gland fails to produce an adequate amount of thyroid hormone to meet metabolic requirements. Early stages of hypothyroidism may not manifest noticeable symptoms. Untreated hypothyroidism might eventually result in various health complications, including cardiovascular issues [4].

Thyroid-stimulating hormone (TSH) concentrations above the reference range and free thyroxine concentrations below the reference range are indicative of overt or clinical primary hypothyroidism. Generally considered to be an indicator of early thyroid failure, mild or subclinical hypothyroidism is characterized by TSH levels above reference levels and free thyroxine levels within normal limits [5,6].

Heart rate variability (HRV) serves as a valuable tool for assessing the autonomic function of cardiovascular activity due to its ability to reflect the dynamic interplay between the two branches of the autonomic nervous system. HRV measures the fluctuation in the time intervals between consecutive heartbeats. Time domain and frequency domain analysis allows for the quantification of the balance between sympathetic and parasympathetic influences on the heart, providing insights into autonomic regulation. Decreased HRV has been demonstrated as a standalone predictor of elevated mortality, serving as an independent risk factor for sudden death, indicative of autonomic dysregulation [7,8,9].

Thyroid hormones play a crucial role in regulating metabolic processes, cardiovascular function, and autonomic nervous system activity. Hypothyroidism can have significant implications for cardiovascular balance due to its effects on multiple physiological processes. Hypothyroidism is a prevalent endocrine illness and has been linked to different cardiovascular abnormalities, including alterations in HRV [1,7].

Our study aims to examine the link between HRV and hypothyroidism and could provide new insights into the relationship between thyroid function and autonomic nervous system regulation. By demonstrating correlations between hypothyroidism and reduced HRV, the study would highlight autonomic dysfunction as a prominent feature of hypothyroidism. This finding would suggest that hypothyroidism is associated with an imbalance between the sympathetic and parasympathetic branches of the autonomic nervous system, contributing to the broader understanding of the systemic impacts of thyroid hormone deficiencies. HRV, as a marker of autonomic imbalance in hypothyroidism, could provide clinicians with a non-invasive diagnostic tool to assess the severity of autonomic dysfunction in these patients and, with further study, could serve as a prognostic indicator, helping to predict the risk of cardiovascular complications associated with hypothyroidism.

## 2. Materials and Methods

### 2.1. Study Population

The study comprises 110 patients with cardiovascular symptoms and hypothyroidism. Patients were initially referred to the cardiology department between January 2023 and December 2023 for cardiovascular symptoms such as palpitations, fatigue, or thoracic constriction. The diagnosis of hypothyroidism was established through clinical evaluation and laboratory testing for thyroid function. None of the patients had a history of thyroid disorder prior to presentation and did not follow any form of thyroid hormone replacement therapy.

Each patient underwent a comprehensive clinical assessment, including a detailed medical history, physical examination, and assessment of cardiovascular symptoms. Data on demographic characteristics, medical history, and thyroid function were collected.

All patients had thyroid function parameters evaluated, including serum levels of thyroid-stimulating hormone (TSH), free thyroxine (FT4), and triiodothyronine (T3) and thyroid autoantibodies.

The elevated blood TSH levels and decreased FT4 levels are signs of hypothyroidism [10]. Serum TSH levels that are slightly elevated (>4.8 mIU/L; normal range: 0.46–4.8 mIU/L) and serum T3 and T4 levels at the bottom of the reference range (FT3: 4.2–8.1 pmol/L, FT4: 10–29 pmol/L) are characteristics of subclinical hypothyroidism [6,10].

Overt hypothyroidism was defined as elevated serum TSH levels (>4.8 mIU/L) and decreased FT4 levels (FT4 < 10 pmol/L). Severe forms of hypothyroidism were defined as TSH levels >10 mIU/L, disregarding FT3 or FT4 levels, and non-severe hypothyroidism refers to both overt and subclinical hypothyroidism with TSH levels < 10 mIU/L. All cases were primary hypothyroidism, and the etiology was both autoimmune and non-autoimmune.

The study excluded patients who had a history of ischemic heart disease, valvular heart disease, brady- or tachy-arrhythmias, atrial fibrillation, heart conduction disorders, branch blocks, arterial hypertension, diabetes mellitus, cerebrovascular disease, chronic obstructive pulmonary disease, liver or renal insufficiency, malignancy, or were taking medication that affects heart rate or heart conduction, such as beta-blockers, non-dihydropyridine calcium channel blockers, or If channel blockers, or other any chronic or acute disease that could affect autonomic function.

The patients had baseline assessments and were reevaluated after 3 months after initiating hormone replacement therapy with therapeutic doses of L-thyroxine as prescribed by an endocrinology specialist.

### 2.2. Heart Rate Variability Analysis

Ambulatory 24-h electrocardiographic (ECG) monitoring was performed for each patient using a 24-h Holter monitor. ECG recordings were analyzed to assess heart rate variability (HRV), arrhythmias, and other electric abnormalities during normal daily activities and sleep, and patients were instructed not to undergo strenuous physical activities. 24-h Holter recordings were obtained at baseline and following 3 months after hormone replacement therapy.

Holter recordings were downloaded onto a computer and analyzed with an ECG Holter program, Labtech Cardiospy. All recordings were also examined visually, and artifacts were deleted manually. All of the recordings had at least 20 h of data after artifact removal.

The HRV parameters were obtained from the Labtech Cardiospy v5.03 software. The HRV parameters used in this study were chosen according to the guidelines of the European Society of Cardiology and the North American Society of Pacemaker and Electrophysiology and used linear method analysis [11].

Linear methods, such as time and frequency domains, are the conventional approaches used to measure HRV. The analysis focused on various parameters related to the time domain, including RR intervals (also known as normal-to-normal intervals or NN intervals), the standard deviation of RR intervals (SDNN), the root mean square of successive RR-interval differences (RMSSDs), and the percentage of adjacent NN intervals varying by more than 50 ms (pNN50). The frequency domain parameters used were TP (total power), low frequency (LF, from 0.04 to 0.15 Hz), and high frequency (HF, from 0.15 to 0.4 Hz). The LF and HF bands can be generated using either the rapid Fourier transform algorithm or autoregressive modeling [12].

Transthoracic echocardiography was performed on all patients. Standard echocardiographic views and measurements were obtained to assess cardiac structure and function, including left ventricular ejection fraction (LVEF). All patients enrolled in the study had a LVEF > 50%.

### 2.3. Statistical Analysis

Descriptive statistics were used to summarize demographic and clinical characteristics of the study population. Data were expressed as means and standard deviations (SDs). Statistical analysis was conducted using MedCalc Version 19.4 and Microsoft Excel 365 (2021). The Student *t*-test was employed for comparing two independent means, while Pearson correlation was used to assess the linear relationship between continuous variables. Significance was set at *p* < 0.05. Analysis of Variance (ANOVA) was conducted to compare HRV parameters between multiple groups, with the Tukey–Kramer test applied for post hoc comparisons to identify specific group differences. To assess the homogeneity of variances, Levene’s test was utilized. Furthermore, the Kolmogorov–Smirnov test was used to assess the normality of data distribution. Confidence intervals (CI) at the 95% level were calculated for HRV parameters to provide an estimate of the precision of the mean values.

The cardiology clinic conducted all of the examinations. All participants provided their informed consent. The study protocol received approval from the ethical committee at our institution.

## 3. Results

### 3.1. Study Group Description

The statistical analysis included a total of 110 patients diagnosed with hypothyroidism. The mean age of the study population was 57.27 + 26.75. In our study cohort, women comprised the majority, accounting for 82.72% of the participants, reflecting the higher prevalence of hypothyroidism in women, which is representative of the disease prevalence. The mean ventricular rate was 77.94 + 12.21. All patients were in sinus rhythm and presented no supraventricular or ventricular arrhythmias or conduction abnormalities on the 24 h ECG monitoring (Table 1).

At baseline, the TSH levels were outside the defined normal range with a mean of 9.89 + 2.81, with FT4 serum levels within normal or lower ranges compared to normal values with FT4 = 13.44 + 4.87. With a significant reduction of TSH value after 3 months of substitution therapy with Levothyroxine, serum TSH levels returned within the normal range with a mean value of 3.65 + 0.74 (95%CI: 5.68 to 6.80; *p* < 0.0001) (Table 2).

All patients were presented with at least one cardiac symptom like palpitations, fatigue, and/or thoracic constriction. We conducted a one-way ANOVA test in order to compare the presence of one or more cardiac symptoms with serum TSH and FT4 values (Table 3 and Table 4).

There was no statistical significance (*p* > 0.05) between the presence of certain cardiac symptoms and the TSH and FT4 serum values. Moreover, between HRV parameters (NN, SDNN, RMSSD, pNN50, TP, HF, and LF) and the symptoms at presentation, there was no significant correlation (*p* > 0.05). After 3 months of substitution therapy, 14 patients still presented with at least one symptom that initially was associated with hypothyroidism.

### 3.2. HRV Analysis

Patients with hypothyroidism after 3 months of substitution therapy with Levothyroxine exhibited improved heart rate variability (*p* < 0.05). After 3 months of treatment, HRV indices obtained from 24 h ECG recordings presented an increase in the NN parameter (mean difference = 334.45; 95%CI: 322.23 to 346.67; *p* < 0.0001) and all parameters that depict time domain analysis: SDNN (mean difference = 30.38; 95%CI: 29 to 31.76; *p* < 0.0001), RMSSD (mean difference = 30.09; 95%CI: 29.12 to 31.05; *p* < 0.0001), and pNN50 (mean difference = 10.62; 95%CI: 10.07 to 11.18; *p* < 0.0001). Regarding the frequency domain parameters, after 3 months of obtaining an euthyroid state, patients exhibit a significant increase in TP (mean difference = 478.71; 95%CI: 465.87 to 491.56; *p* < 0.0001), HF amplitude (mean difference = 140; 95%CI: 135.53 to 145.48; *p* < 0.0001), a slight increase in LF amplitude (mean difference = 8.87; 95%CI: 5.35 to 12.38; *p* < 0.0001), with a decrease in the LF/HF ratio (mean difference = −0.96; 95%CI: −1.05 to −0.87; *p* < 0.0001) (Table 5).

We used Pearson correlation (r: Pearson correlation coefficient) to analyze the relationship between TSH levels and HRV parameters. In hypothyroid patients, there is a significant correlation between TSH value and HRV parameters, with a moderate inverse relationship between TSH and SDNN (r = −0.3849, 95%CI = −0.53 to −0.21, *p* < 0.0001), a strong inverse relationship between TSH and RMSSD (r = −0.4917, 95%CI = −0.62 to −0.33, *p* < 0.0001), TSH and pNN50 (r = −0.6404, 95%CI = −0.73 to −0.51, *p* < 0.0001), TP (r = −0.4618, 95%CI = −0.59 to −0.30, *p* < 0.0001), and HF (r = −0.6773, 95%CI = −0.76 to −0.56, *p* < 0.0001). There was no statistically significant relationship between TSH levels and the LF parameter (*p* = 0.7371). Moreover, there is a positive relationship between TSH and LF/HF ratio (r = 0.5932, 95%CI = 0.45 to 0.70, *p* < 0.0001). The significant inverse relationships between TSH and most HRV parameters (SDNN, RMSSD, pNN50, TP, HF) suggest that higher TSH levels are associated with decreased HRV, which indicates reduced vagal (parasympathetic) activity. This aligns with the suppression of vagal activity noted in hypothyroid patients. The positive correlation between TSH and LF/HF ratio indicates that as TSH levels increase, the balance shifts more towards sympathetic dominance. This suggests an improvement in sympathovagal balance after treatment. Overall, these findings imply that in hypothyroid patients, elevated TSH levels are linked with reduced vagal activity and a shift towards sympathetic dominance, which is reflected in the HRV parameters (Table 6).

We conducted a comparative analysis of HRV parameters between patients with severe hypothyroidism and those with non-severe hypothyroidism at baseline and after 3 months of treatment. Most parameters showed no significant difference in variances, except for pNN50, TP, HF (*p* < 0.001), and LF/HF (*p* = 0.023) at baseline, indicating different variances for these parameters between severe hypothyroidism and non-severe. Significant differences (*p* < 0.001) were found for SDNN, RMSSD, pNN50, TP, and HF at both baseline and 3 months. LF, representing both sympathetic and parasympathetic activity, was not significantly different between the groups at baseline and after 3 months. The LF/HF ratio, an indicator of autonomic balance, was significantly higher in the severe group at baseline (*p* < 0.001) and after 3 months (*p* < 0.001), indicating a dominance of sympathetic over parasympathetic activity in the severe group. NN showed no significant differences, indicating similar means between the groups. Severe hypothyroidism patients had significantly different HRV parameters compared to non-severe patients, with the severe group generally having lower values for most parameters (*p* < 0.05). Severe hypothyroidism patients typically had lower HRV measures compared to non-severe patients, indicating lower autonomic nervous system activity in the severe group (Table 7).

## 4. Discussion

The autonomic regulation of physiological functions is mediated by the sympathetic and parasympathetic nervous systems. Elevated sympathetic activity can precipitate symptoms such as accelerated heart rate and elevated blood pressure during periods of relaxation. Conversely, manifestations like heart rate variability and orthostatic hypotension may indicate dysfunction within the parasympathetic system. Hypothyroidism poses an increased risk of cardiovascular mortality and morbidity [13,14].

The main objective was achieved, indicating a link between thyroid dysfunction and autonomic dysfunction, as evidenced by the analysis of heart rate variability. Individuals with hypothyroidism exhibited a reduction in HRV. This phenomenon could potentially be attributed to the adverse impact of TSH. The rise in sympathetic activity and decline in parasympathetic function may carry clinical significance [15,16].

Various mechanisms have been proposed to elucidate the autonomic disturbances observed in thyroid disorders. These include heightened plasma adrenaline levels with concomitant receptor or post-receptor desensitization, diminished chronotropic responses to β-adrenergic stimulation, and direct influences on sympathetic outflow and cardiac responsiveness to thyroid hormone. Moreover, hypothyroidism is associated with escalated production, release, and plasma degradation of catecholamines, consequently augmenting sympathetic activity [13,14,15].

Similarly, the observed decline in parasympathetic activity in hypothyroidism may be attributable to alterations within cardiac parasympathetic neurons, resulting in decreased muscarinic effects. These collective insights underscore the intricate interplay between thyroid function and autonomic regulation, highlighting potential avenues for therapeutic intervention in managing cardiovascular complications associated with hypothyroidism [13,15,17].

During rest, the RR interval changes continuously around the average value. Conventionally, the interrelationship between sympathetic and vagal modulation on the heart is considered reciprocal; the increase of activity in one system causes the decrease of activity in the other.

The vegetative balance with its two sympathetic and parasympathetic branches is illustrated by the RR variability components. In the time domain, the SDNN, RMSSD, and pNN50 are considered as indices of vagal activity. In the frequency domain, the high-frequency component is mainly modulated by the parasympathetic nervous system and corresponds to the parasympathetic tone, and the low-frequency component is under the influence of both sympathetic and parasympathetic systems [3,11].

The suppression of vagal activity surpasses the escalation of sympathetic activity in intensity. The reduction in HRV primarily stems from a significant decline in vagal tone. TSH functions as a neurotransmitter and stimulates sympathetic output from the central nervous system, thus exerting a pivotal influence on the sympathovagal balance. This observation supports the notion of a more pronounced decrease in HRV among patients with elevated TSH levels [1,17].

In our study, as well as in the Galleta et al. study [18,19], there were notable correlations between thyroid function parameters and heart rate variability (HRV) measures, particularly in hypothyroid patients. Galleta et al. found a negative association between SDNN and serum TSH levels in subclinical hypothyroid patients, while the LF/HF ratio showed a positive correlation with TSH levels. Similarly, in our study, we observed a decrease in HRV among patients with higher TSH levels, suggesting a link between thyroid dysfunction and autonomic imbalance. Moreover, Galleta et al. reported lower HRV measures in patients with overt hypothyroidism compared to controls, indicating a reduction in autonomic function in these individuals. This finding aligns with our observation of decreased HRV in hypothyroid patients in our study cohort [18,19].

Furthermore, both studies demonstrated improvements in HRV parameters following treatment with Levothyroxine (L-T4) in hypothyroid patients. Galleta et al. [18,19] reported that HRV measures became comparable to those of control subjects after six months of L-T4 treatment, indicating the restoration of autonomic function with the correction of hypothyroidism. Similarly, our study found that HRV parameters improved after treatment, suggesting that thyroid hormone replacement therapy may positively impact autonomic regulation in hypothyroid patients.

Hiremath et al. [20] observed that hypothyroid patients exhibited alterations in HRV parameters indicative of reduced parasympathetic and increased sympathetic activity. Specifically, lower values of HF power were associated with diminished vagal modulation of the heart, while higher values of LF norm suggested heightened sympathetic activity. The LF/HF ratio, considered a marker of sympathovagal balance, indicated an altered sympathovagal balance in hypothyroid patients. These conclusions align with our study’s findings regarding the autonomic imbalance observed in hypothyroidism [20,21].

Overall, the findings from our study are consistent with some literature studies [1,18,19], providing further evidence of the association between thyroid dysfunction and alterations in autonomic function, as well as the beneficial effects of L-T4 treatment on HRV in hypothyroidism.

Brusseau et al. [1] synthesize findings from various studies comparing HRV between hypothyroid patients and healthy controls. There is consistent observation of altered HRV parameters in hypothyroid patients compared to healthy individuals with a reduction in parasympathetic activity and an increase in sympathetic activity among hypothyroid patients. Our study is in accordance with the general medical literature [1].

Moreover, in our study, patients with severe hypothyroidism exhibited lower HRV compared to those with non-severe hypothyroidism, indicating reduced autonomic nervous system activity in the severe group. Severe hypothyroidism significantly affected HRV parameters, resulting in lower values for SDNN, RMSSD, pNN50, TP, and HF. These reductions underscore a more pronounced decline in autonomic nervous system function in patients with severe hypothyroidism, indicating a direct relationship between the severity of the disease and the autonomic imbalance. Despite observed improvements after 3 months of treatment, significant differences in HRV parameters between the severe and non-severe groups persisted. This indicates persistent autonomic dysfunction, with lower parasympathetic activity and higher sympathetic dominance as reflected by the LF/HF ratio. These reductions underscore a more pronounced decline in autonomic nervous system function in patients with severe hypothyroidism, indicating a direct relationship between the severity of the disease and the autonomic imbalance.

In our study, we observed that vagal inhibition exhibits greater intensity compared to increased sympathetic activity, resulting in a more significant reduction in high-frequency (HF) power as opposed to low-frequency (LF) power. The clinical implications of hypothyroidism include reduced vagal tone and increased sympathetic activity [16]. A higher risk of coronary artery disease and cardiovascular death is linked to hypothyroidism. A sympathovagal imbalance is one of the several processes causing these complications.

There are few limitations to our study. Extended electrocardiogram (ECG) recordings can be prone to artifacts and interference, especially during ambulatory monitoring. Unlike short-term recordings, ambulatory monitoring takes place in a less controlled environment, potentially introducing more challenges in maintaining signal quality. Nevertheless, ambulatory monitoring has the benefit of recording parameter values over an extended period of time, providing a more thorough evaluation of HRV and autonomic function. Moreover, while the study included a higher proportion of women, which is representative of the disease prevalence, this demographic distribution should be considered when interpreting the results.

## 5. Conclusions

In conclusion, our study underscores the association between hypothyroidism and decreased heart rate variability (HRV). The decrease in HRV in hypothyroid individuals can be explained by intricate molecular mechanisms that involve catecholamines, as well as the impact of thyroid-stimulating hormone (TSH) on the regulation of HRV. These findings highlight the relationship between thyroid function and autonomic regulation, with implications for cardiovascular health and possible clinical management. The sympathovagal tone reduction observed in hypothyroidism highlights the significance of addressing autonomic dysfunction in this population.

## Figures and Tables

**Table 1 diagnostics-14-01261-t001:** Characteristics of the study group.

Description	Values *
Age (years)	57.27 + 26.75
Sex (M/F)	19/91
HR (bpm)	77.94 + 12.21
SBP (mmHg)	124.20 + 8.65
DBP (mmHg)	77.65 + 3.89
TSH (mIU/L) baseline	9.89 + 2.81
FT4 baseline	13.44 + 4.87
Severe/non-severe hypothyroidism	49/61

HR = heart rate; SBP = systolic blood pressure; DBP = diastolic blood pressure. * Values are presented as mean + SD for continuous variables and as absolute frequency for categorical variables.

**Table 2 diagnostics-14-01261-t002:** TSH reduction after 3 months of Levothyroxine replacement therapy.

Description	Values
Difference	−6.24
Standard error	0.27
Standard deviation	2.91
95% CI ^1^	−6.80 to −5.68
t-statistics	−22.02
DF ^2^	109
Significance level	*p* < 0.0001

^1^ Confidence interval. ^2^ Degrees of freedom.

**Table 3 diagnostics-14-01261-t003:** Relationship between symptoms and TSH level.

Symptoms *	*n*	Mean	SD
1	23	10	2.66
2	12	9.93	3.12
3	15	9.24	2.47
4	20	9.73	3.10
5	15	9.35	1.92
6	12	10.26	3.27
7	13	10.93	3.34

* Symptoms: 1 = palpitations; 2 = fatigue; 3 = thoracic constriction; 4 = palpitations and fatigue; 5 = palpitations and thoracic constriction; 6 = fatigue and thoracic constriction; 7 = palpitations, fatigue, and thoracic constriction; *n* = number of patients; mean = mean TSH value; SD = standard deviation.

**Table 4 diagnostics-14-01261-t004:** Relationship between symptoms and FT4 level.

Symptoms *	*n*	Mean	SD
1	23	13.33	3.89
2	12	11.89	3.98
3	15	15.41	5.32
4	20	13.56	6.23
5	15	13.65	4.75
6	12	12.97	4.65
7	13	12.83	4.97

* Symptoms: 1 = palpitations; 2 = fatigue; 3 = thoracic constriction; 4 = palpitations and fatigue; 5 = palpitations and thoracic constriction; 6 = fatigue and thoracic constriction; 7 = palpitations, fatigue, and thoracic constriction; *n* = number of patients; mean = mean FT4 value; SD = standard deviation.

**Table 5 diagnostics-14-01261-t005:** HRV parameters at baseline and after 3 months of substitution therapy.

*n* = 110	H		H3	
	Mean	SD	Mean	SD
NN	935.43 *	41.30	1269.89	50.11
SDNN	107.85 *	13.63	138.23	14.84
RMSSD	35.05 *	6.98	65.14	8.86
pNN50	11.23 *	2.94	21.86	4.28
TP	2669.20 *	334.69	3147. 92	337.22
HF	327.24 *	85.21	467.75	89.36
LF	936.31 *	156.56	945.19	159.89
LF/HF	3.05 *	0.92	2.08	0.50

*n*: number of patients; H: parameters at baseline with hypothyroidism; H3: parameters after 3 months of substitution treatment; SD: standard deviation; NN: normal-to-normal intervals; SDNN: standard deviations of all RR intervals; RMSSD: root mean square of successive difference of RR; pNN50 (%): percent of successive RR differences > 50 ms; TP: total power; HF: high frequency power; LF: low frequency power; LF/HF: low frequency/high frequency ratio, * *p* < 0.0001.

**Table 6 diagnostics-14-01261-t006:** Correlation between TSH value and HRV parameters.

TSH	SDNN	RMSSD	pNN50	TP	HF	LF	LF/HF
r	−0.3849	−0.4917	−0.6404	−0.4618	−0.6773	−0.0323	0.5932
95% CI	−0.53 to −0.21	−0.62 to −0.33	−0.73 to −0.51	−0.59 to −0.30	−0.76 to −0.56	−0.21 to 0.15	0.45 to 0.70
*p*	<0.0001	<0.0001	<0.0001	<0.0001	<0.0001	0.7371	<0.0001

TSH: Thyroid-stimulating hormone; SDNN: standard deviation of all RR intervals; RMSSD: root mean square of successive difference of RR; pNN50 (%): percent of successive RR differences > 50 ms; TP: total power; HF: high frequency power; LF: low frequency power; LF/HF: low frequency/high frequency ratio; r: Pearson correlation coefficient; 95%CI: confidence interval for r.

**Table 7 diagnostics-14-01261-t007:** Comparative analysis of heart rate variability parameters between severe and non-severe hypothyroidism patients at baseline and after 3 months of treatment.

Parameter	Baseline			After 3 Months		
Groups	Severe *	Non-Severe *	P1	Severe *	Non-Severe *	P1
NN	931.27 (40.34)	938.79 (42.10)	*p* = 0.345	1269.59 (45.84)	1270.13 (53.68)	*p* = 0.956
SDNN	101.66 (7.47)	112.82 (15.39)	*p* < 0.001	132.50 (10.29)	142.84 (16.34)	*p* < 0.001
RMSSD	30.37 (6.15)	38.82 (5.09)	*p* < 0.001	60.86 (8.90)	68.59 (7.23)	*p* < 0.001
pNN50	8.88 (1.34)	13.13 (2.47)	*p* < 0.001	19.40 (3.28)	23.85 (3.98)	*p* < 0.001
TP	2477.94 (214.58)	2822.85 (335.85)	*p* < 0.001	2959.02 (233.49)	3299.67 (332.30)	*p* < 0.001
HF	251.37 (40.02)	388.20 (58.59)	*p* < 0.001	389.33 (43.69)	530.75 (62.64)	*p* < 0.001
LF	908.06 (156.52)	959.02 (154.14)	*p* = 0.090	915.49 (159.52)	969.05 (157.46)	*p* = 0.081
LF/HF	3.70 (0.86)	2.54 (0.60)	*p* < 0.001	2.38 (0.48)	1.86 (0.39)	*p* < 0.001

NN: normal-to-normal intervals; SDNN: standard deviation of all RR intervals; RMSSD: root mean square of successive difference of RR; pNN50 (%): percent of successive RR differences > 50 ms; TP: total power; HF: high frequency power; LF: low frequency power; LF/HF: low frequency/high frequency ratio. 1ANOVA between groups; * mean (standard deviation).

## Data Availability

Data are contained within the article.

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
