# Peer review of "Hypothyroidism and Heart Rate Variability: Implications for Cardiac Autonomic Regulation"

_diagnostics, 2024, doi:10.3390/diagnostics14121261_

Round 1

Reviewer 1 Report

Comments and Suggestions for Authors

Reviewer 2 Report

Comments and Suggestions for Authors

The authors of the manuscript present results from an original study, including 110 patients with hypothyroidism, in which they investigate heart rate variability. Authors' hypothesis is that thyroid dysfunction (in this study - hypothyroidism) could exert influence on regulation of cardiac autonomic function and myocardial performance. The manuscript is topical. Hypothyroidism is very common among the general population of many countries. It is mostly due to Hashimoto thyroiditis, radioactive iodine therapy or surgery (for hyperthyroidism, goiter, or thyroid cancer),  iodine deficiency. Cardiovascular complications of overt hypothyroidism (myxedema) have been described a long time ago, however there is lack of sufficient information about the long-term effect of milder forms of hypothyroidism on the cardiovascular function, including on the cardiac autonomic function and myocardial performance.

I have the following recommendations and questions to the authors:

1. I recommend modification of the title so that it be more concise, for example "Hypothyroidism and Heart Rate Variability: Implications for Cardiac Autonomic Regulation".

2. Style of some citation numbers in the text must be corrected (2 or more citations should be included in the same brackets, but not separately). 

3. There is no control group of participants, matched by gender, age and other characteristics, except for the thyroid function. 

4. Are all included patients with new-diagnosed hypothyroidism? Or some patients have been already diagnosed with hypofunction (and some of them on therapy) at the time of inclusion?

5. Tables must be improved. 

6. It is not quite clear from the results, if compensation of thyroid function has improved HRV and regulation of cardiac autonomic function.

7. Authors mention about myocardial performance at the end of Abstract. "Myocardial performance" should be removed, although it would be a valuable result (if really present in results). Investigation of "myocardial performance" requires at least echocardiography (including tissue Doppler imaging and myocardial strain/strain rate). 

8. Discussion must be improved. Authors must begin it with discussing their own results ("Our results showed....", or "In our study we found...."), comparing them to publications of other authors: what was similar, what was different, what remains unclear, what has to be done in future investigations...

Comments on the Quality of English Language

English language is generally fine, just minor stylistic correction are necessary. 

Reviewer 3 Report

Comments and Suggestions for Authors

The paper adds only a larger patient number to the two studies by Galetta F, cited in the references.

It would have been interesting to present separate analysis for overt and subclinical hypothyroidism.

Lines 38-40: in a paper for a medical review, the definition of autonomic system is redundant.

Lines 46-48 and 50-52: also the definition of hypothyroidism appears redundant.

Line 71: the number of patients enrolled should appear in the Results section, not in the Methods.

Lines 71-75: the text should state clearly if patients were selected for symptoms and then hypothyroidism was diagnosed (in this case, how many with symptoms without hypothyroidism) or for the presence of hypothyroidism with symptoms.

Line 82: this sentence interacts with the previous note. Inclusion criteria should be stated clearly.

Line 113: “calculated by computer” is not a clear statement. Did the Authors use the previously mentioned Labtech Cardiospy program?

Line 131: descriptive statistics used should be listed in this section. The mean±1 SD used for continuous variables should be described here (line 135?) and not in the Results section after any use. If different statistics are to be used in non-normal variables, how was deviation from Gaussian distribution assessed?

Line 134: Pearson’s correlation does not perform a comparison between groups.

Line 143 and following: such a high number of decimals is irrelevant, and probably one in enough.

Line 143: “Age distribution was higher in women (82.72%)”. The meaning of this sentence is obscure and different age values in men and women (and heir comparison) do not appear in the text.

Line 147: TSH values out of normal range was an inclusion criterion.

Lines 147-152: the sentence is obscure. The value in line 151 appears to be the TSH level after treatment, but the confidence interval refers to the variation of TSH.

Table 1: standard deviation and not standard error should be reported. The 218 degrees of freedom indicate that an unpaired t-test was performed, while this is a typical situation for paired t-test. It should also be noted that the confidence interval is not symmetrical around the average value.

Tables 2 and 3: with such a high number of symptom class and a relative low number of patients, it is really difficult to evidence any difference. In any case the title is misleading, as no relationship is assessed.

Lines 169-171: this sentence belongs to the next paragraph. In any case, how can symptoms, with a categorical classification, be correlated with HRV quantitative values?

Line 174: The meaning of “regarding the severity” is obscure. Do the Authors mean independently of the severity? But if this is the case, neither data nor comparison are presented.

Line 177: if this is the confidence interval of the variation in the parameter, also the average value should be reported (in the table?). In fact in table 4 the indication of the statistical significance with the star, and moreover close to the baseline values, is not clear for the reader.

Table 4: the decimal separator is here comma, while it is period in the remaining tables and in the text.

Line 181: total power is not listed in the Method section among the HRV parameter evaluated.

Line 196: table 5 is mentioned in the text, but it is not present.

References: the format of the references is not according to “Diagnostics” rules

Comments on the Quality of English Language

The quality of English is rather good and it needs only a minor revision.

Round 2

Reviewer 1 Report

Comments and Suggestions for Authors

The authors have addressed my previous comments. I have only a few minor suggestions for this revised version.

First, could you please clarify if the definitions of "severe" and "non-severe" hypothyroidism are the same as "severe forms" and "overt hypothyroidism" mentioned in lines 99 to 101?

Second, it seems that some periods (i.e., the decimal point indicator) are missing from the numbers in Table 5.

Overall, I am supportive of the publication of this article after these minor revisions are made.

Reviewer 2 Report

Comments and Suggestions for Authors

The authors have observed most of the reviewer's recommendations and have significantly improved the quality of their manuscript.

There are still some minor technical errors (like spaces between the citation indexes in the brackets and a double space between "7" and the following bracket in line 262) that could be easily fixed.  After correcting these technical errors the manuscript can be published in its current version. 

Reviewer 3 Report

Comments and Suggestions for Authors

Line 83-85 -  It is still not clear if patients were recruited according to the presence of cardiovascular symptoms, ad then those who presented hypothyroidism were included in the study. Even if Authors in the accompanying letter state that “We did not collect data about the patients that presented with cardiac symptoms and were not diagnosed with a thyroid disorder.” It would be interesting to know the percentage of symptomatic patients with hypothyroidism.

Line 144 - The sentence “Comparisons between groups were performed using several appropriate statistical tests” is useless.

Line 150, 164 etc – in the description of continuous variables is defined in the Statistical Analysis section, it is useless to repeat mean+SD for each variable.

Line 163 - as the study enrolls only patients with hypothyroidism, it is redundant to repeat “in patients with hypothyroidism”

Table 1: the second column heading should not be “Mean+SD” as absolute frequencies also are reported

Line 170 and following - was the deviation of these variables from Gaussian distribution tested with Kolmogorov-Smirnov test, as stated in the Statistical Analysis section?

Line 175 - the abbreviation “CI” is not defined in the text.

Table 2 - mean difference and confidence interval should have the same sign (below zero to indicate a reduction of TSH).

Line 197 - to which comparison is this statistical significance referred?

Line 199 - average value of the increase should be reported, not only confidence interval; moreover, if the increase is reported, there is no need to specify “of the difference”.

Lines 215-216 - the text in parentheses is not necessary.

Lines 215-224 and table 6 – the two limits of the confidence interval should be separated by “to” like in the remaining of the text.

Line 222 - a not significant correlation should not be described as a “weak relationship” as no proof of any relationship is obtained.

Lines 242-245 – no need to repeat a statement already appearing in the Statistical Analysis section.

Lines 239 and following: if the comparison of interest is between severe and non-severe hypothyroidism, there is no need to use ANOVA, and a simple unpaired T-test would be enough; moreover, T-test is rather robust to heteroscedasticity. With two groups only, the use of Tukey-Kramer test is meaningless. Even the meaning of the use of Kolmogorov-Smirnov test on residual is not clear.

Table 7 Is difficult to read; it would be better to report baseline and 3-month data in parallel column, with a column for variation and with statistical comparison for each group. The ANOVA was performed separately for baseline and 3-month data and do not inform on the different variations.

References still do not follow standard format.

Comments on the Quality of English Language

Only minor editing required
